# Influence of Elective Cesarean Calving (with and without Dexamethasone Induction) on the Erythrogram and Iron Serum Profiles in Nellore Calves

**DOI:** 10.3390/ani12121561

**Published:** 2022-06-17

**Authors:** Luan Ricci Silva, Renan Braga Paiano, Mariana Guimarães de Oliveira Diogo, Melina Marie Yasuoka, Ana Claúdia Birali, Mayara Berto Massuda, Maria Luiza Kuhne Celestino, Daniela Becker Birgel, Flávio José Minieri Marchese, Paulo Fantinato Neto, Vanessa Martins Storillo, Eduardo Harry Birgel Junior

**Affiliations:** 1Department of Anatomy of Domestic and Wild Animals, School of Veterinary Medicine and Animal Sciences, University of São Paulo, 87 Professor Orlando Marques de Paiva Avenue, São Paulo 05508-010, Brazil; luanricci@gmail.com (L.R.S.); renanpaiano@usp.br (R.B.P.); melinamarie@gmail.com (M.M.Y.); flaviominierivet@hotmail.com (F.J.M.M.); fantinato@usp.br (P.F.N.); veterinariavanessa@yahoo.com.br (V.M.S.); ehbirgel@usp.br (E.H.B.J.); 2Department of Animal Reproduction, School of Veterinary Medicine and Animal Sciences, University of São Paulo, 87 Professor Orlando Marques de Paiva Avenue, São Paulo 05508-010, Brazil; 3Department of Veterinary Medicine, College of Animal Science and Food Engineering, University of São Paulo, 225 Duque de Caxias, Pirassununga 13635-900, Brazil; bilariana@gmail.com (A.C.B.); ma.massuda@gmail.com (M.B.M.); mkcelestino@gmail.com (M.L.K.C.); dabirgel@usp.br (D.B.B.)

**Keywords:** hematology, elective cesarean section, Nellore calves, neonatal phase

## Abstract

**Simple Summary:**

The use of elective cesarean section has become common practice in double muscled breeds of in vitro fertilized cattle and cloned calves, particularly before the onset of labor, when it is known that cesarean section negatively affects neonatal respiration and metabolic adaptation in humans and calves. However, there is a lack of information on the effects of cesarean section on the erythrogram of calves. The objectives of this study are to characterize the hematological profile of Nellore calves born spontaneously or by elective C-section (with or without induction with dexamethasone) and to verify the frequency of anemia in these animals. Our data indicate that in the first day of life there was a decrease in the number of red blood cells, hemoglobin rates, and values of the globular volume regardless of the type of birth; however, the recovery of these hematological values happened faster in zebu calves born spontaneously than in the group born by elective C-section (with or without induction). The results suggest that iron supplementation in the first month of life in Nellore calves that underwent cesarean section could be recommended to prevent the iron deficiency anemia observed in this study.

**Abstract:**

The aim of the present study is to evaluate the erythrogram and iron serum profiles of neonatal calves born spontaneously or born by elective cesarean section with or without dexamethasone induction. The research was performed on 38 newborn Nellore calves. Three groups of calves were assigned according to the type of birth: calves born by spontaneous vaginal calving (*n* = 10), calves born by elective cesarean section without inducing labor (*n* = 14), and calves born by elective cesarean section with labor induction with dexamethasone (*n* = 14). Blood samples to assess red blood cell count (RBC), hemoglobin, hematocrit, mean corpuscular volume (MCV), mean corpuscular hemoglobin (MCH), concentration of mean corpuscular hemoglobin (MCHC), serum iron (SFe), total capacity to bind iron to transferrin (TIBIC), and transferrin saturation index (TSI) were performed at calving (0, 3, 6, and 12 h of life) and on 1, 2, 3, 5, 7, 10, 15, and 30 days of life. Regardless of the experimental group (calves born spontaneously, or born by elective cesarean section with or without dexamethasone induction), in the first day of life there was a decrease in the number of red blood cells, hemoglobin rates, and values of the globular volume. In the period of the first 10 days of life, animals from spontaneous vaginal delivery quickly recovered values of erythrocytes, hemoglobin, and packed cell volume, whereas animals born by elective C-section (induced and uninduced) did not recover as quickly in their rates of hemoglobin and packed cell volume values. In calves born by elective C-section (induced and uninduced), it was observed in their period between 10 and 30 days of life that the MCV and MCH were reduced by passing the presenting microcytic hypochromic when compared with calves obtained by spontaneous vaginal delivery. In the period between 10 and 30 days of life, the levels of SFe and TSI in animals born by elective C-section (induced and uninduced) are significantly lower. The differences in the erythrogram values between Nellore calves born spontaneously and those by elective C-section with or without induction must be considered consequent to the process of neonatal adaptation to extrauterine life. Iron supplementation in the first month of life in calves from cesarean could be recommended to prevent anemia of this iron deficiency.

## 1. Introduction

Brazil has a commercial herd of 187.55 million head of cattle, thus having great potential in the meat market and, placing first in the ranking of the largest beef exporters [1]. Estimates suggest that about 80% of the Brazilian national cattle herd for meat production is Nellore or Nellore crossbred [2]. The Nellore breed is a descendant of the Ongole Indian herd, a common animal in the old Madras Province, located on the east coast of India. Most of these animals were imported to Brazil from India from the 1960s onwards and stood out in Brazil for their hardiness and better productivity in relation to the different breeds already present in the country [3]. In the late 1960s, the implementation of zootechnical tests for the selection of genetically superior animals began, and since then, the selection of more productive animals has been intensified so that it has been possible to reduce the age at the slaughter of commercial herds [4,5].

The use of reproductive biotechnologies, including artificial insemination, embryo transfer, and cloning, contribute to an improvement in the genetics of herds, which may increase meat and milk production, improve fertility rates, and increase animal productivity [6,7]. Some health disorders, however, can occur due to the intensive use of biotechnologies [8]. One of these changes is the increase in calf weight at birth, which can increase the incidence of dystocia and the need for caesarean sections [9,10].

The cesarean section aims to guarantee the survival of the fetus and the cow, which also guarantees the future fertility of the mother. Indications for performing this surgical procedure may be of maternal or fetal origin, such as very young heifers, pelvic deformities, failure to dilate the cervix, uterine torsion, large and poorly positioned fetuses, and prolonged gestation [11]. The use of elective C-section without an initial attempt at a natural birth has become common practice in double-muscled breeds of cattle such as the Belgian Blue [12]. In animals from in vitro embryo production (IVP) and especially in cloned calves, it has been recommended to perform elective C-sections as a result of prolonged pregnancies and not signaling births in these animals [13].

Calves born by elective C-section may have less vitality due to the increased risk of developing metabolic disorders and respiratory diseases in the first moments of life [12]. The elective C-section delivery affects the expression of key genes involved in the efficiency of the pulmonary liquid to air transition at birth, and may lead to an increased inflammatory response in jejunal tissue, which could compromise colostral immunoglobulin absorption in calves [14]. The large offspring syndrome is associated with respiratory distress syndrome, and carbohydrate metabolism disorders were observed in cloned Nellore calves. The occurrence of moderate to severe normocytic and normochromic anemia was reported in cloned Nellore calves born by elective C-section, with the anemia gradually setting in from 12 h of life and reaching a maximum intensity at the end of the first week, then a gradual recovery of values occurs from the 15th day onwards [15].

Dexamethasone is a glucocorticoid similar to cortisol, which can be used as a labor inducer of calving, as it decreases plasma levels of progesterone from the 8th day of administration; moreover, the use of corticosteroids can contribute to a greater efficiency in the pulmonary maturation of neonates, reducing the risk of death in the first moments of the newborn’s life [16,17,18].

The evaluation of blood tests is essential for the early diagnosis of diseases in newborn animals by contributing to a rapid and efficient intervention in neonates, increasing the survival rate of animals in the first month of life, and reducing the economic loss due to death of the animals. Although there is abundant literature on blood tests in calves around the world [19,20,21,22,23], we found little information about the erythrogram profile in Nellore calves. Thus, the objective of the present study is to characterize the hematological profiles of Nellore calves born either spontaneously or by elective C-section (with or without induction with dexamethasone) and to verify the frequency of anemia in these animals and the iron serum profiles.

## 2. Materials and Methods

### 2.1. Animals

The study was conducted at the School of Animal Husbandry and Food Engineering in the University of São Paulo, Pirassununga, Brazil. In total, 38 healthy newborn female Nellore calves that were conceived by artificial insemination were used in this study. During the experimental period, the animals were bottle-fed with four liters of milk, twice a day, and they had free access to hay and water. All areas of the study was performed with the approval of the Bioethics Committee of the School of Veterinary Medicine and Animal Sciences, University of São Paulo, São Paulo, Brazil (Protocol No. 2291110714/2016).

In this prospective observational study, three groups of calves were assigned according to their type of birth. One group consisted of calves born by spontaneous vaginal calving (*n* = 10); a second group included calves born by elective C-section, without inducing labor (*n* = 14), and the third group included calves born by elective C-section, with labor induction (*n* = 14) performed by applying dexamethasone (Cortiflan^®^, OuroFino Animal Health, São Paulo, Brazil; 25 mg, I. M.) 24 h before the surgical procedure. Calves were born between 282 and 292 days of gestation. Calves that had dystocic birth, or that were not born between 282 and 292 days of gestation or that had any clinical disease were excluded the study.

### 2.2. Sampling

Blood samples were taken by puncture of the external jugular vein at birth (0, 3, 6, and 12 h of life) and on 1, 2, 3, 5, 7, 10, 15, and 30 days of life. For the hematological analysis, blood samples were collected in vacuum glass tubes of 5 mL containing ethylenediaminetetraacetic acid (EDTA) as an anticoagulant (Vacutainer^®^ Systems, Becton Dickinson, Franklin Lakes, NJ, USA). For iron serum profiles, the blood samples were collected into a tube without anticoagulant (Becton Dickinson Vacutainer^®^ Systems, Franklin Lakes, NJ, USA). Blood samples were transported to the laboratory and analyzed within 3 h of collection. In the laboratory, these samples were centrifuged at 2000× *g* for 15 min. The supernatant was transferred to a sterile plastic test tube and were stored at −20 °C until analysis. Analyses of red blood cell count (RBC), hemoglobin, hematocrit, mean corpuscular volume (MCV), mean corpuscular hemoglobin (MCH), and concentration of mean corpuscular hemoglobin (MCHC) were performed using a veterinary hematology analyzer (BC-2800 Vet Mindray^®^) with species specific settings for cattle.

The determination of serum iron content and total capacity to bind iron to transferrin (TIBIC) was performed using a colorimetric method (commercial kit SI250 of the Randox^®^) by the automatic biochemical analyzer RX Dayton^®^-Randox Laboratories, Crumlin, United Kingdom. The transferrin saturation index (TSI) was obtained by calculating the ratio between the iron concentration and Total Transferrin Iron Binding Capacity (TIBIC) and multiplied by 100.

### 2.3. Statistical Methods

Statistical analyses were performed according to SAS (SAS^®^, version 9.3 SAS/STAT; SAS Institute Inc., Cary, NC USA). Erythrogram and iron serum profiles were analyzed by the mixed-model procedure for repeated measurements (PROC MIXED) of SAS. The model included the variable experimental group as a fixed effect and the variable and calf as a random effect. Mean values were compared with the Tukey’s test. Differences with *p* < 0.05 were considered significant. Values are presented as the mean ± standard error (SE).

## 3. Results

The study was carried out with 38 healthy Nellore female calves conceived by artificial insemination that were born between 282 and 292 days of gestation. One group consisted of calves from spontaneous vaginal delivery (*n* = 10); the second group consisted of calves born by elective C-section without the induction of parturition (*n* = 14), and the third group included calves from elective C-section with the induction of calving with dexamethasone application. Calves that had calving problems or were sick were excluded from the study. During the first 24 h of life period and regardless of the experimental group, there was a decrease in the number of red blood cells (*p* < 0.05), hemoglobin rates (*p* < 0.05), and values of the hematocrit (*p* < 0.05). During this period, there was an increase in the percentage of anemic animals from 2.63% (1/38) at birth to 26.32% (10/38) at 24 h of life. Calves were diagnosed with anemia based on values of the erythrogram; the disorder was characterized by a red blood cell (RBC) count, hemoglobin (Hb) concentration, and hematocrit (Ht) value lower than 5.0 × 10^6^/μL, 8.0 g/dL, and 24%, respectively, as recommended in the literature [24]. Calves that had RBC, Hb, and Ht values > 5.0 × 10^6^/μL, >8.0 g/dL and >24% were classified as non-anaemic.

At 10 days of life, animals from spontaneous vaginal delivery were observed to be making a relatively quick recovery in the values of hemoglobin and packed cell volume, whereas in animals born by elective C-section (induced and uninduced) had slower recovery in the rates of hemoglobin and packed cell volume values (Figure 1A–C and Appendix A).

At 10 and 15 days of life, calves born by elective C-section without the induction of labor had lower Hb and Ht values than those born by elective C-section with the induction of labor with dexamethasone. In these moments it was observed that the frequency of anemia was higher in calves born without induced birth, respectively, 42.85% (6/14) and 35.71% (5/14) than the observed in calves born by dexamethasone-induced delivery, 7.14% (1/14) and 7.14% (1/14).

Calves born spontaneously were found to recover more quickly from this anemia than calves obtained by elective C-section. At 30 days of life, no calf born spontaneously was anemic, while 14.28% (2/14) of calves born with induction and 25.57% (4/14) of calves born by elective C-section without induction of birth were anemic.

Based on the values of the hematimetric indices, the calves born by spontaneous vaginal calving had higher (*p* < 0.05) values of MCV in the periods of 7, 10, 15, and 30 days of life than the calves born by induced and non-induced elective C-section and had higher (*p* < 0.05) values of MCH in the periods of 10 and 15 days of life than the calves born by induced and non-induced elective C-section (Figure 1E,F and Appendix A).

The serum iron values in the first hours of life (groups 0 h and 3 h of life) were lower in calves born by spontaneous vaginal calving than that observed in calves born by elective C-section (with and without labor induction). Between 6 h and 1 day of age, no difference was observed in serum iron values between the groups; from 2 days of age, calves born by spontaneous vaginal calving had an increase in serum iron values. Calves born by spontaneous vaginal calving had higher (*p* < 0.05) values of serum iron in the periods of 2, 3, 5, 7, 10, 15, and 30 days of life than the calves born by induced and non-induced elective C-section and had higher (*p* < 0.05) values of TIBIC in the periods of 12 h, 1, 2, 3, 5, 7, 10, 15, and 30 days of life than the calves born by induced and non-induced elective C-section (Figure 2A,B and Appendix A). Transferin saturation index (TSI) represents the ratio of serum iron to TIBIC. Calves born by spontaneous vaginal calving had lower TSI (*p* < 0.05) at birth, with 3 and 12 h hours of life than the calves born by non-induced elective C-section and had higher values of TSI (*p* < 0.05) in the periods of 2, 3, 5, 10, 15, and 30 days of life than the calves born by non-induced elective C-section (Figure 2C and Appendix A).

## 4. Discussion

This study provides an understanding of the erythrogram of newborn calves born under different conditions. The results described in the present study for the red blood cell, hemoglobin and hematocrit are in line with those described by Biondo et al., who studied the influence of age on the hemogram of healthy Nellore calves. The lowest values of the red blood cell, hemoglobin, and hematocrit were observed in the first day of age of the calves [25]. The reduction in erythrogram values during the first week after birth was previously described and may occur due to several factors, the main one being the expansion of plasma volume after the ingestion of colostrum, which, due to its high protein concentration, can contribute to an increase in oncotic pressure causing dilution of the plasma [19,26,27,28]. Other related factors are that the destruction of fetal hematietric and less bone marrow activity in the newborn may contribute to a lower erythropoiesis [28]. These changes can contribute to the development of physiological anemia during this phase, which recover to normal values within the first 30 days of life [29].

Our results demonstrated an increase in the percentage of anemic animals from 2.63% (1/38) at birth to 26.32% (10/38) in the first day of life. In that time, 30.0% (3/10) of calves born spontaneously were anemic, while 14.28% (2/14) of calves born with induction and 35,71% (5/14) of calves born by elective C-section without induction of parturition were anemic.

Anemia is a cause of morbidity and mortality in neonates, but the frequency of its occurrence is variable [30]. In Jersey and Holstein cows in the United States, 15.8% of the animals had this condition [31]. In Iranian herds, the percentage of anemic Holstein calves was 17.7% [32]. In Brazilian herds it was reported that 14.29% of Holstein calves [29] and 30.7% of crossbred calves with the five-day-old animals were anemic [29,33]. In Holstein calves, it was that observed that 78.2% of the cases of anemia occurred within 14 days of birth (40% between birth and seven days of birth and 38.2% in the second week of life) [29]. In our results, we observed the same: Birth has a great influence on the occurrence of anemia, which is more frequent in the first week of life. These hematological changes in the first week of life are not necessarily indicative of disease but reflect the physiological variations that occur during the transitional period. Although anemia in calves has a multifactorial etiology, iron deficiency was considered the most frequently observed determining cause of anemia [29,31,34].

The influence of the cesarean section on the erythrogram of newborn calves has been little studied in Buiatrics [35,36]. It has been demonstrated that Holstein calves born with obstetric assistance (traction or cesarean section) presented a decrease in the number of red blood cells, in the hemoglobin rate, and in the globular volume 12 h after birth [35]. While other researchers described that the number of red blood cells, hematocrit, and hemoglobin concentrations found in newborn Belgian Blue calves born by elective C-section were higher than those observed in Holstein calves born by spontaneous vaginal delivery without assistance [36]. However, no research has studied the blood pattern of Zebu breeds submitted to cesarean section.

In our research, differences were observed at 10 days of life in the erythrograms of calves obtained by elective C-section. The recovery of red blood cell, hemoglobin and hematocrit values observed in the group of Zebu calves born spontaneously was faster than in the group born by elective C-section with or without induction. This rapid recovery in the erythrogram values may favor the immunological status of these animals, and may contribute to a lower incidence of disease during this phase, favoring greater weight gain during the weaning phase.

Based on the values of the hematimetric indices, the results of MCV and MCH exhibit a gradual decrease in their values at birth to up to 30 days of life in groups of calves born by elective C-section with or without induction, which may be related to the replacement of fetal hemoglobin by adult hemoglobin during this phase [37,38].

At 2 days of life, our results demonstrated that iron levels in animals obtained by elective C-section were lower in serum iron than those in spontaneous vaginal delivery. In human babies, the impact of cesarean section with iron deficiency has been studied [39]. When the newborn’s umbilical cord is clamped and cut immediately after cesarean section, about 20 mL of all the baby’s blood remains in the placenta; this has a high potential that affects iron stores and can put the baby at risk for anemia [40]. Late umbilical cord screening is considered a strategy to improve ferritin levels at birth and to prevent childhood anemia. Better hemoglobin levels in infants with late clamping (clamped after the cord pulsations ceased) were proven in research in the field of human medicine and that the process of milking the umbilical cord at the time of the cesarean brings benefits in relation to anemia in babies [41,42].

The process of anemia in the newborn is also due to the often uterine or vaginal compression when compared to normal delivery; therefore, the newborns’ pulmonary fluids are not eliminated efficiently during the delivery process, which can consequently delay the onset of breathing and prevent blood transfusion from the placenta [43]. It was also observed that the TIBIC, which indirectly represents the amount of transferrin, is lower in Nellore calves obtained by elective C-section. Respiratory Distress Syndrome associated with caesarean section may determine pulmonary inflammation and tissue hypoxia [44]. Inflammatory conditions, which enable the sequestration of iron, could explain the iron deficiency [29]. In human babies, respiratory discomfort associated with cesarean section determines a decrease in the serum concentration of transferrin in the blood [45]. One of the hypotheses of our research was that inflammation resulting from acute respiratory distress syndrome could determine the release of cytokines that could interfere with hematopoiesis and thus be associated with a greater occurrence of anemia observed on group the animals with elective c-section without induction. Erythropoiesis may be impaired due to inflammatory disease, mainly due to the action of inflammatory cytokines that can cause direct toxicity in erythroid precursor cells [46]. In chronic inflammatory diseases, the proliferation and differentiation of erythroid precursors are impaired because of an altered response to erythropoietin [47,48]. While these findings are interesting, they should be taken with caution due to the relatively small number of calves used in this study.

## 5. Conclusions

Our data indicate that in the first day of life there was a decrease in the number of red blood cells, hemoglobin rates, and values of the globular volume, regardless of the type of birth. The recovery of these hematological values happened faster in Zebu calves born spontaneously than in the group born by elective C-section with or without induction; in addition, at two days of life of the calves, iron levels were lower than those obtained by elective C-section compared to spontaneous vaginal delivery. The differences in the erythrogram values of Nellore calves born spontaneously or by elective C-section with or without induction, must be considered consequent to the process of neonatal adaptation to extrauterine life. Iron supplementation in the first month of life in calves undergoing C-section, or even dystocia, could be recommended to prevent this iron deficiency anemia.

## Figures and Tables

**Figure 1 animals-12-01561-f001:**
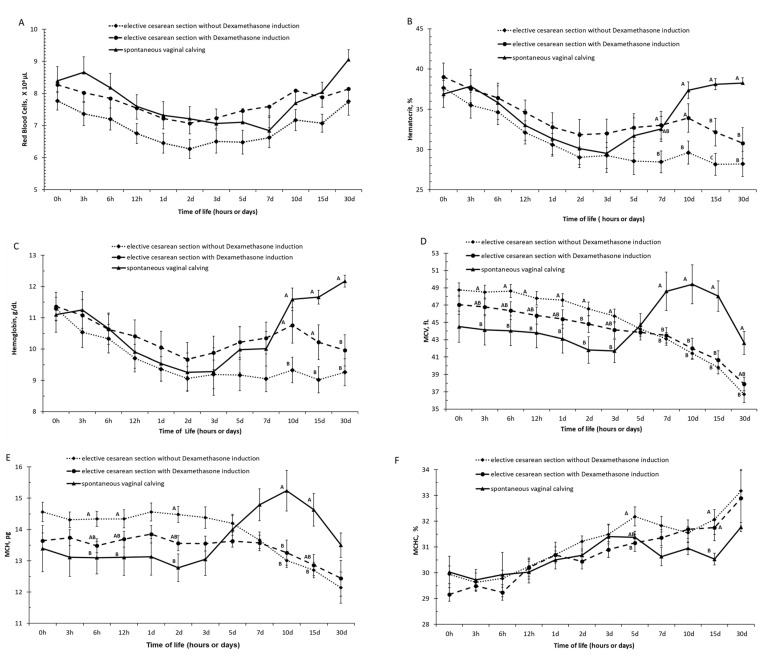
(**A**) Red blood cells, (**B**) hematocrit, (**C**) hemoglobin, (**D**) mean corpuscular volume (MCV), (**E**) mean corpuscular hemoglobin (MCH), and (**F**) concentration of mean corpuscular hemoglobin (MCHC) in calves born by spontaneous vaginal calving (*n* = 10) and elective cesarean calving with (*n* = 14) or without (*n* = 14) induction during the first month of life. A, B, C—Different letters on the line mean statistically significant difference compared with Tukey’s teste, at 5% significance.

**Figure 2 animals-12-01561-f002:**
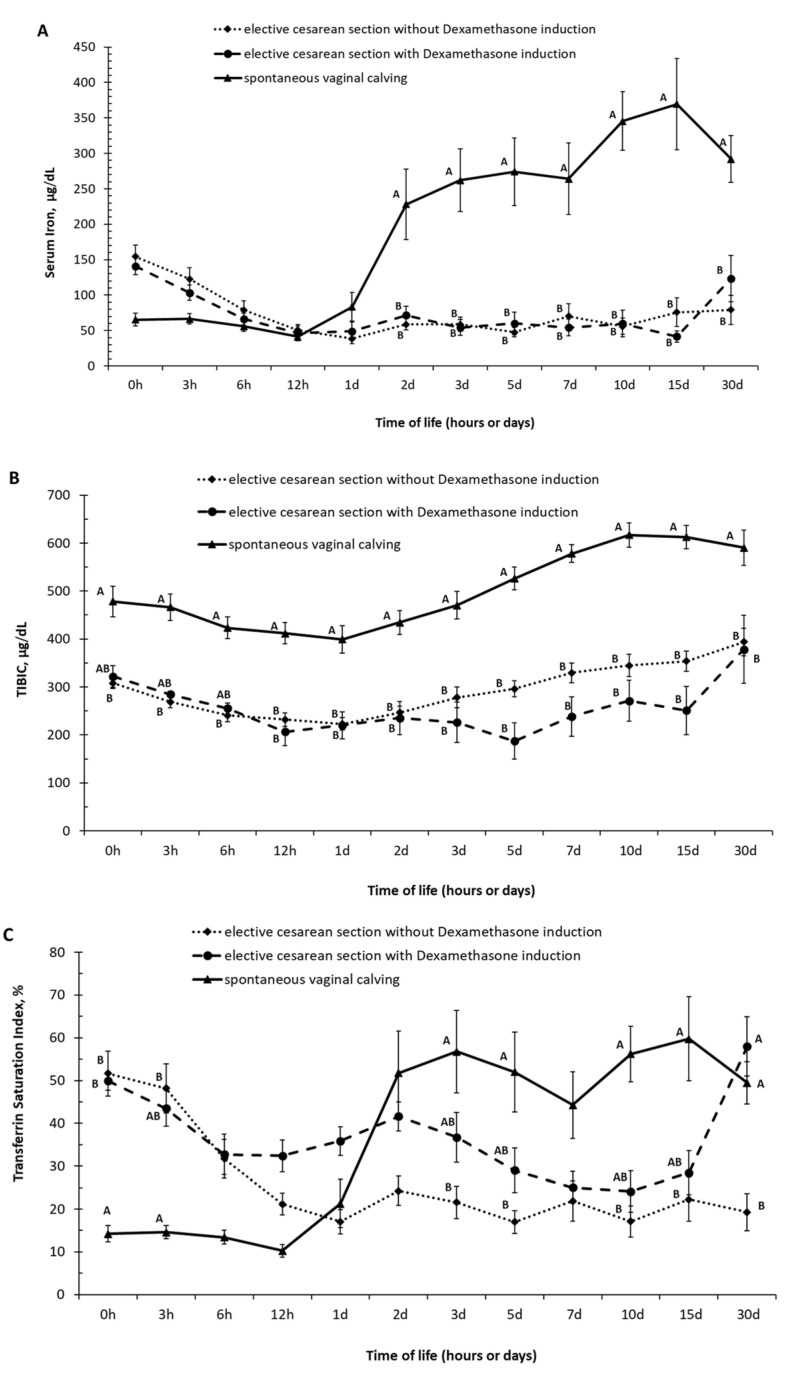
(**A**) Serum iron, (**B**) total capacity to bind iron to transferrin—TIBIC, (**C**) transferin saturation index—(TSI, (means ± SEM) in calves born by spontaneous vaginal calving (*n* = 10) and elective cesarean calving with (*n* = 14) or without (*n* = 14) induction during the first month of life. A, B, C–Different letters on the line mean statistically significant difference compared with Tukey’s teste, at 5% significance.

## Data Availability

The corresponding author can provide the data that support the findings of this study upon request.

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
