# Peer review of "Influence of Elective Cesarean Calving (with and without Dexamethasone Induction) on the Erythrogram and Iron Serum Profiles in Nellore Calves"

_animals, 2022, doi:10.3390/ani12121561_

Round 1

Reviewer 1 Report

The authors have made the proposed changes and therefore I believe the paper is now suitable for publication.

Reviewer 2 Report

See attached file.

Author Response

This manuscript is a resubmission of an earlier submission. The following is a list of the peer review reports and author responses from that submission.

Round 1

Reviewer 1 Report

The paper by Dr Silva et colleagues is an interesting experimental trial with three groups of calves (control and two treatments groups) that investigates the effects of cesarean section (CS) and the dexamethasone (dex) injection on the erythrogram of calves.

My major concern is that the authors don’t present clearly their results as concern the effect of dexamethasone. So I would propose at the results they present the relations in the investigated parameters between the CS group and CS-dex group, and also at the discussion to discuss the effect of dex on the erythrogram.

In addition I have one correction:

Lines 120-121. The sentence ‘Calves that had RBC, Ht and Hb values > 5.0 × 106 /μl, > 8.0 g/dL and > 24% were classified as non-anaemic. ’ should be ‘Calves that had RBC, Hb and Ht values > 5.0 × 106 /μl, > 8.0 g/dL and > 24% 120 were classified as non-anaemic. ’
